

# Mean wind vector estimation using the Velocity-Azimuth-Display (VAD) method: An explicit algebraic solution

Gerd Teschke[1] and Volker Lehmann[2]

[1]Institute for Computational Mathematics in Science and Technology, University of Applied Sciences Neubrandenburg, Brodaer Str. 2, 17033 Neubrandenburg
[2]Deutscher Wetterdienst, Meteorologisches Observatorium Lindenberg, Am Observatorium 12, 15848 Tauche-Lindenberg

*Correspondence to:* Gerd Teschke (teschke@hs-nb.de)

**Abstract.** This paper deals with the analysis of the sampling setup for Doppler profilers aiming at the determination of vertical profiles of the wind. An explicit solution for the retrieval of mean wind vectors under the assumption of local homogeneity is presented for the case of a symmetric Velocity-Azimuth-Display sampling and a stability analysis is performed. Furthermore, the explicit solution allows a detailed investigation of the propagation of radial wind measurement errors on the retrieved wind
vector.

## 1   Introduction

The wind vector is a fundamental variable to describe the state of the atmosphere (Dutton, 1986); it can be measured with a variety of in-situ and remote sensors. The latter are typically ground-based active systems emitting artificially generated electromagnetic (Lidar, Radar) or acoustic waves (Sodar). In the so-called Doppler systems, the frequency shift of the scattered

waves is used to measure the motion of the scattering medium directly. While the details of the measurement process differ between the various instruments, the common feature of all Doppler instruments is that the velocity can only be determined along a single direction, called the line-of-sight or radial direction. This provides merely one component of the full 3D wind vector. Of course it is possible to use three Doppler instruments to sample the same volume from different directions, but this approach is impractical for operational meteorology (Stephens, 1994) and only used for special applications (Fuertes et al.,

2014). Vertical wind profiling attempts to estimate the wind vector as a function of height using data from a single instrument.

A number of different retrieval methods have been proposed for uniform or linear wind fields (Browning and Wexler, 1968; Waldteufel and Corbin, 1979; Koscielny et al., 1984; Caya and Zawadzki, 1992; Stephens, 1994). These methods are known as Velocity-Azimuth-Display (VAD), Volume Velocity Processing (VVP) or Doppler Beam-Swinging (DBS) and its different variants are successfully used in operational meteorology.

It is obvious that such simplifying assumptions can in general not be made for the instantaneous wind field in a turbulent flow. However, it is customary to decompose a turbulent wind field into a mean and a fluctuating component describing the deviations from the mean (Salby, 1996; Davidson, 2004; Vallis, 2006). This is justified by the claim of an existing spectral gap between the mean flow and (microscale) turbulence (Stull, 1988). Indeed, evidence for such a spectral gap the boundary layer was recently reported by Larsén et al. (2016). The wind measurement can then be split up in two tasks, namely first





the determination of the mean value and second, the estimation of the Reynolds stress tensor or other statistical parameters describing the turbulent part of the flow (Sathe and Mann, 2013; Sathe et al., 2015; Newman et al., 2016).

For operational applications, like data assimilation for numerical weather prediction models, the interest is clearly in the mean wind. This is due to the fact that such models are unable to resolve the small (turbulent) scales directly. Processes on
such scales must instead be parameterized (Warner, 2011). For operational Doppler profilers it is therefore enough to aim at the determination of the mean wind vector profile, with typical averaging times of O(10 min). This restriction makes it more likely, that simplifying assumptions like homogeneity or linearity of the wind field, hold at least on average without incurring large errors. In fact the assumption of statistical (local) homogeneity and quasi-steadiness can often be applied in boundary layer meteorology (Wyngaard, 2010) even though it is clear that there are limits to these assumptions (Maurer et al., 2016).
Practical experiences from comparisons of various wind retrieval methods with Doppler radars suggest that the simplest methods for the retrieval of the horizontal winds give the best accuracy in comparison with independent wind sensors (Holleman, 2005) - a seemingly counterintuitive result, given the large area scanned in comparison with special wind profiling instruments like radar wind profilers or Doppler lidars (Cifelli et al., 1996). A possible explanation is that the retrievals using more complex wind field models are ill-conditioned (Shenghui et al., 2014). Given these results and the importance for
wind profile measurements for operational meteorology, it seems therefore appropriate to further investigate the wind retrieval methods for the rather simple assumption of horizontal homogeneity and quasi-steadiness (or stationarity).

The paper is organized as follows: The first section is concerned with an algebraic description of the wind retrieval problem for a Doppler profiler. This includes an extension/recasting to a frame-based sensing concept. In the second section, an explicit solution for the case of symmetric VAD sampling with constant elevation is provided. This allows a direct calculation of the
retrieval error and provides a guideline for an optimal sampling configuration.

## 2   Reconstruction of constant wind vector

Under the assumption of a stationary, horizontally homogeneous and vertically piecewise constant wind field, the wind retrieval can be described algebraically. The new aspect is an interpretation of the sensing setup based on the mathematical concept of frames (Christensen, 2008) which allows, for specific sensing scenarios, an explicit computation of involved matrices and
therewith an explicit derivation of the associated eigenvalues.

### 2.1   Sensing model and retrieval

For a given azimuth $\alpha$ and zenith distance $\phi$, the beam direction can be described by a unit vector given as

$$
\boldsymbol{e} = \begin{pmatrix} \sin\alpha\sin\phi \\ \cos\alpha\sin\phi \\ \cos\phi \end{pmatrix} \in \mathbb{R}^3 \,,
$$

The goal is to retrieve an unknown wind vector $\boldsymbol{v} \in \mathbb{R}^3$ from projections of $\boldsymbol{v}$ on a set of different beam vectors $\{\boldsymbol{e}_k\}_{k=1}^N$.
This set of beam vectors defines the spatial sampling. The assumption is that within the sampling volume and sampling





time, the wind vector to be determined $\boldsymbol{v} \in \mathbb{R}^3$ is constant, i.e. within the sampling volume we assume a constant wind. This assumption appears to be overly restrictive, but the goal is not to determine the instantaneous wind vector in an arbitrary turbulent wind field, buth rather the mean (horizontal) wind vector over an averaging time of $O(10-30\ min)$. For the average wind field, horizontal homogeneity has to be assumed over the area spanned by the beam directions, which is for Doppler

profilers typically $O(0.1-10\ km)$ and stationarity has to be assumed over the averaging time. In the vertical, the wind field is assumed to be piecewise constant over layers with a thickness of the order of the radial resolution of the Doppler profiler, namely $O(10-100\ m)$.

The sampling process can be described through the application of projection matrices. Geometrically the projection of a vector $\boldsymbol{v}$ onto a vector $\boldsymbol{e}$ can be described by the following $3 \times 3$ matrix $P = \boldsymbol{e}\boldsymbol{e}^T$, which easily follows since the projection

of $\boldsymbol{v}$ onto $\boldsymbol{e}$ can be expressed as $\langle \boldsymbol{e}, \boldsymbol{v} \rangle \boldsymbol{e} = \boldsymbol{e} \langle \boldsymbol{e}, \boldsymbol{v} \rangle = \boldsymbol{e}\boldsymbol{e}^T \boldsymbol{v} = P\boldsymbol{v}$, where $\langle \cdot, \cdot \rangle$ denotes the inner product, i.e. for two given vectors $\boldsymbol{a}, \boldsymbol{b} \in \mathbb{R}^3$ we have $\langle \boldsymbol{a}, \boldsymbol{b} \rangle = \sum_{i=1}^3 = a_i \cdot b_i = \boldsymbol{a}^T \boldsymbol{b}$. Note that the magnitude of $\boldsymbol{p}$ is equal to the (radial) component of the wind field in the beam direction. By construction, the projection $P$ is a rank one matrix, and, moreover, $P$ is idempotent and symmetric, i.e. $PP = P$ and $P^T = P$. Assume now, that the spatial sampling consists of $N$ beam vectors, $\boldsymbol{e}_1, \ldots, \boldsymbol{e}_N$, for which we can associate $N$ projections, $P_1, \ldots, P_N$. Each beam direction provides us with one radial velocity vector, denoted

by $\boldsymbol{p}_k$, $k = 1, \ldots, N$, hence for each $k$ we can write a $3 \times 3$ linear system $\boldsymbol{p}_k = P_k \boldsymbol{v}$. Combining all $N$ linear systems into one single system results in

$$
\begin{pmatrix} \boldsymbol{p}_1 \\ \vdots \\ \boldsymbol{p}_N \end{pmatrix} = \underbrace{\begin{pmatrix} P_1 \\ \vdots \\ P_N \end{pmatrix}}_{P} \boldsymbol{v} \, , \tag{1}
$$
$$
\underbrace{\phantom{\begin{pmatrix} \boldsymbol{p}_1 \\ \vdots \\ \boldsymbol{p}_N \end{pmatrix}}}_{\boldsymbol{p}}
$$

where $\boldsymbol{p} \in \mathbb{R}^{3N}$ and $P \in \mathbb{R}^{3N,3}$. As each $P_k$ is of rank one and as we are usually faced with noisy measurements, directly solving (1) is impossible. A stable retrieval of $\boldsymbol{v}$ can be achieved through a minimization of $\|\boldsymbol{p} - P\boldsymbol{v}\|^2$ with respect to $\boldsymbol{v}$. The

optimal $\boldsymbol{v}$ is given through the solution of the normal equation,

$$
(P^T P)\boldsymbol{v} = P^T \boldsymbol{p} \quad . \tag{2}
$$

A unique solution requires invertibility of $P^T P$ in (2), which can be achieved if the rank of $P^T P$ equals three. Hence, at least three linear independent beam directions are (obviously) required to obtain a unique solution. To obtain feasible numerical approximations of $\boldsymbol{v}$, one has to ensure numerical stability of the inversion process especially in the case of noisy data, i.e. we

have to ensure reasonable approximation quality also for the case $\boldsymbol{p}^\epsilon = \boldsymbol{p} + \boldsymbol{\epsilon}$ with $\|\boldsymbol{\epsilon}\| \le \delta$. As we have to solve the normal equation, we first express the symmetric map $P^T P$ by its eigensystem, $P^T P = U D U^T$, where $U$ is the orthogonal matrix of eigenvectors of $P^T P$ and $D = \mathrm{diag}(\lambda_1, \lambda_2, \lambda_3)$ is the diagonal matrix of eigenvalues of $P^T P$. Then, it follows that

$$
\boldsymbol{v} = (P^T P)^{-1} P^T \boldsymbol{p} = U D^{-1} U^T P^T \boldsymbol{p} \, . \tag{3}
$$

With $\boldsymbol{v}^\epsilon := (P^T P)^{-1} P^T \boldsymbol{p}^\epsilon$, we obtain

$$
\|\boldsymbol{v} - \boldsymbol{v}^\epsilon\| \le \|(P^T P)^{-1} P^T (\boldsymbol{p} - \boldsymbol{p}^\epsilon)\| \le \|U D^{-1} U^T P^T\| \|\boldsymbol{\epsilon}\| \le \|U\| \|D^{-1}\| \|U^T\| \|P^T\| \delta \le \frac{\delta}{\lambda_{\min}}, \tag{4}
$$



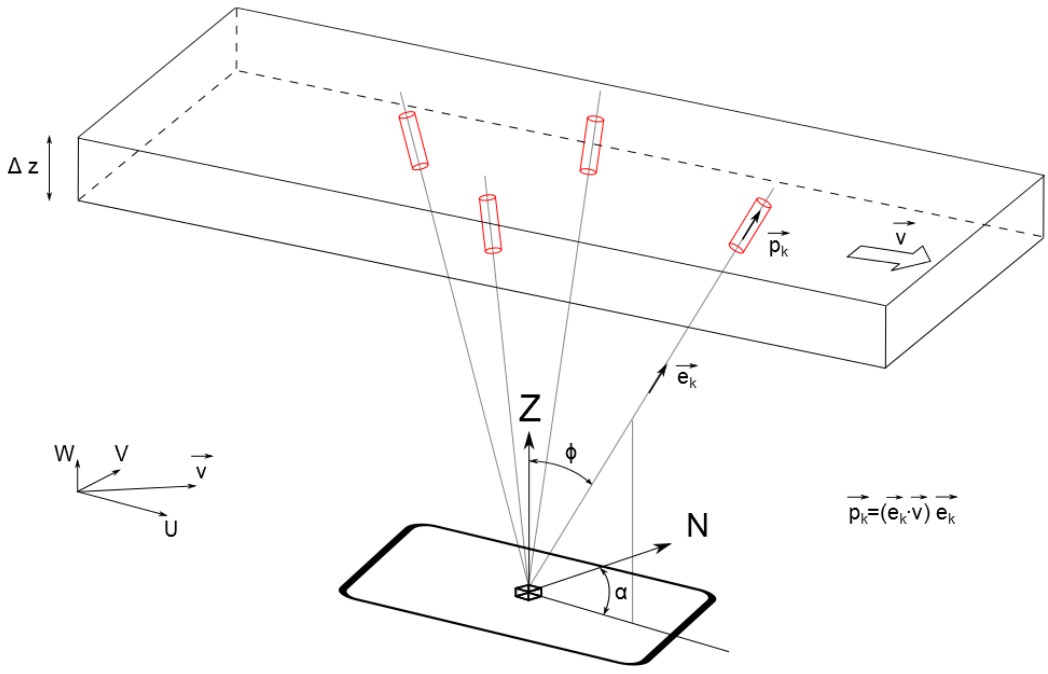

**Figure 1.** Schematics of sampling for $N = 4$.

where $\lambda_{\min}$ denotes the smallest eigenvalue. Therefore, the recovery error can be minimized by maximizing the smallest eigenvalue of $P^T P$. This can be achieved by a proper choice of the corresponding beam vectors $e_1, \ldots, e_N$. Hence, the main question to answer is how to set-up the beam vectors determining the spatial sampling.

## 2.2 Frame-based recast of the sampling design

In order to answer this question, we consider the set of beam vectors as a frame. Without mathematical rigor, a frame can be seen as a collection of vectors that span the full vector space and which are not necessarily linear independent. Such a system of vectors is called overcomplete or redundant and allows to represent given vectors in different ways (non-uniqueness). The redundancy has useful error suppressing effects. Using this approach, the goal is to find a simple description of reconstruction stability and reconstruction error in dependence on the sampling design.

A set of vectors $\{e_k\}_{k=1}^N$ forms a frame for $\mathbb{R}^3$ if there exist constants $0 < A \leq B < \infty$, the so-called frame bounds, such that for all $v \in \mathbb{R}^3$,

$$A\|v\|^2 \leq \sum_{k=1}^N |\langle v, e_k \rangle|^2 \leq B\|v\|^2 \quad \text{or equivalently,} \quad \langle Av, v \rangle \leq \langle Sv, v \rangle \leq \langle Bv, v \rangle, \tag{5}$$





where $S$ is the frame operator introduced below. This frame condition ensures first, that all radial components have finite energy and second, that the set of beam directions is complete, i.e. there exists no (wind) vector in $\mathbb{R}^3$ that is orthogonal to all beam directions.

Let us first reformulate the reconstruction problem. Let $\boldsymbol{e}_1, \dots, \boldsymbol{e}_N \in \mathbb{R}^3$ denote the individual unit vectors of beam directions and consider the so-called pre-frame operator $T : \mathbb{R}^N \to \mathbb{R}^3$, $T\boldsymbol{c} = \sum_{k=1}^N c_k \boldsymbol{e}_k$ , with adjoint, $T^* : \mathbb{R}^3 \to \mathbb{R}^N$, given by $T^* = \{\langle \cdot, \boldsymbol{e}_k \rangle\}_{k=1}^N$. Then the frame operator defined as $S = TT^* : \mathbb{R}^3 \to \mathbb{R}^3$, is given by

$$S = \sum_{k=1}^N \langle \cdot, \boldsymbol{e}_k \rangle \boldsymbol{e}_k \ , \tag{6}$$

which is self-adjoint and symmetric. The frame operator (6) relates to the above mentioned projections as follows,

$$S = TT^* = P_1 + \dots + P_N = P_1 P_1 + \dots + P_N P_N = P_1^T P_1 + \dots + P_N^T P_N = P^T P \ , \tag{7}$$

and thus the invertibility of $S$ is ensured by selecting three linear independent projections (as already mentioned). In what follows we aim to elaborate how the number of beam directions might change the frame bounds of $S$, which coincide with the smallest and largest Eigenvalues of $P^T P$, i.e. for the bounds in (5) we have $A = \lambda_{\min}$ and $B = \lambda_{\max}$.

In order to provide an explicit computation of the solution and therewith an explicit stability analysis, we recast the optimization problem by means of the pre-frame operator $T$, i.e. we aim to find an equivalent formulation for $\|\boldsymbol{p} - P\boldsymbol{v}\|_{\mathbb{R}^{3N}}^2$. First, we have $T^* = (\boldsymbol{e}_1, \dots, \boldsymbol{e}_N)^T : \mathbb{R}^3 \to \mathbb{R}^N$, and by $\boldsymbol{p}_k = \boldsymbol{e}_k V_k = \boldsymbol{e}_k (\boldsymbol{e}_k)^T \boldsymbol{v} = P_k \boldsymbol{v}$ and by $(T^*)^T = T$, the normal equation reads as

$$S\boldsymbol{v} = (T^*)^T T^* \boldsymbol{v} = (T^*)^T \boldsymbol{V} \ , \tag{8}$$

where $\boldsymbol{V} \in \mathbb{R}^N$ is comprised of the radial wind components for the given beam configuration, see e.g. Päschke et al. (2015). This holds true due to

$$P^T P \boldsymbol{v} = S\boldsymbol{v} = \sum_{k=1}^N \langle \boldsymbol{v}, \boldsymbol{e}_k \rangle \boldsymbol{e}_k = \begin{pmatrix} \boldsymbol{e}_1 & \dots & \boldsymbol{e}_N \end{pmatrix} \begin{pmatrix} (\boldsymbol{e}_1)^T \\ \vdots \\ (\boldsymbol{e}_N)^T \end{pmatrix} \boldsymbol{v} = (T^*)^T T^* \boldsymbol{v} = (T^*)^T \boldsymbol{V} = P^T \boldsymbol{p} \ . \tag{9}$$

The equivalence of the optimization problems is immediate,

$$
\begin{aligned}
\|\boldsymbol{p} - P\boldsymbol{v}\|_{\mathbb{R}^{3N}}^2 &= \left\| \begin{pmatrix} \boldsymbol{p}_1 \\ \vdots \\ \boldsymbol{p}_N \end{pmatrix} - \begin{pmatrix} \boldsymbol{e}_1 (\boldsymbol{e}_1)^T \boldsymbol{v} \\ \vdots \\ \boldsymbol{e}_N (\boldsymbol{e}_N)^T \boldsymbol{v} \end{pmatrix} \right\|_{\mathbb{R}^{3N}}^2 = \sum_{k=1}^N \|\boldsymbol{p}_k - \boldsymbol{e}_k (\boldsymbol{e}_k)^T \boldsymbol{v}\|_{\mathbb{R}^3}^2 = \sum_{k=1}^N \|\boldsymbol{e}_k V_k - \boldsymbol{e}_k (\boldsymbol{e}_k)^T \boldsymbol{v}\|_{\mathbb{R}^3}^2 \\
&= \sum_{k=1}^N \|\boldsymbol{e}_k\|_{\mathbb{R}^3}^2 \left( V_k - (\boldsymbol{e}_k)^T \boldsymbol{v} \right)^2 = \sum_{k=1}^N \left( V_k - (\boldsymbol{e}_k)^T \boldsymbol{v} \right)^2 = \|\boldsymbol{V} - T^* \boldsymbol{v}\|_{\mathbb{R}^N}^2 \ ,
\end{aligned}
$$

and we have a reduction of dimension by a factor three.





## 3 Explicit solution and error analysis

In practice, the linear system (3) can be solved numerically through the Singular Value Decomposition, see e.g. Päschke et al. (2015), to minimize errors from finite computational accuracy. This method provides numerical solutions in the general case and it can therefore be implemented in operational Doppler systems. Nevertheless, an explicit solution of (3) would provide

more insight into error propagation and thus allow a further investigation of optimal sampling conditions. Such an explicit solution can indeed be given for a VAD-like sampling scenario. In the following section it is shown that all the involved quantities and error bounds can be explicitly calculated. As these error bounds depend directly on the sensing parameters, the sampling design can be optimized towards a minimal error in the retrieval.

### 3.1 Equispaced circular VAD-like sampling

With preassigned equispaced azimuth angles $\alpha_k = 2\pi k/N$, $k = 0, \ldots, N-1$ and constant zenith distance $\phi$ we have

$$T^* = \begin{pmatrix} \sin\alpha_0 \sin\phi & \cos\alpha_0 \sin\phi & \cos\phi \\ \vdots & & \\ \sin\alpha_{N-1}\sin\phi & \cos\alpha_{N-1}\sin\phi & \cos\phi \end{pmatrix}, \ \boldsymbol{V} = \begin{pmatrix} V_0 \\ \vdots \\ V_{N-1} \end{pmatrix}, \ \boldsymbol{v} = \begin{pmatrix} u \\ v \\ w \end{pmatrix}. \tag{10}$$

The minimization of $\|\boldsymbol{V} - T^*\boldsymbol{v}\|^2$ results in $S\boldsymbol{v} = T\boldsymbol{V}$ or, equivalently, in $P^TP\boldsymbol{v} = P^T\boldsymbol{p}$. Hence, in order to provide an explicit expression for the solution of this linear system, we have to derive $P^TP$, which is given by,

$$P^TP = \begin{pmatrix} \sum_{k=0}^{N-1}\sin^2\alpha_k\sin^2\phi & \sum_{k=0}^{N-1}\sin\alpha_k\cos\alpha_k\sin^2\phi & \sum_{k=0}^{N-1}\sin\alpha_k\sin\phi\cos\phi \\ \sum_{k=0}^{N-1}\sin\alpha_k\cos\alpha_k\sin^2\phi & \sum_{k=0}^{N-1}\cos^2\alpha_k\sin^2\phi & \sum_{k=0}^{N-1}\cos\alpha_k\sin\phi\cos\phi \\ \sum_{k=0}^{N-1}\sin\alpha_k\sin\phi\cos\phi & \sum_{k=0}^{N-1}\cos\alpha_k\sin\phi\cos\phi & \sum_{k=0}^{N-1}\cos^2\phi \end{pmatrix}. \tag{11}$$

The key for evaluating this matrix is interpreting each of the entries as finite geometric series. Therefore, with the help of the following summations,

$$\sum_{k=0}^{N-1}\sin^2\alpha_k = \sum_{k=0}^{N-1}\left(\frac{e^{i\alpha_k}-e^{-i\alpha_k}}{2i}\right)^2 = \frac{N}{2}$$

$$\sum_{k=0}^{N-1}\cos^2\alpha_k = \sum_{k=0}^{N-1}\left(\frac{e^{i\alpha_k}+e^{-i\alpha_k}}{2}\right)^2 = \frac{N}{2}$$

$$\sum_{k=0}^{N-1}\sin\alpha_k = \sum_{k=0}^{N-1}\frac{e^{i\alpha_k}-e^{-i\alpha_k}}{2i} = 0$$

$$\sum_{k=0}^{N-1}\cos\alpha_k = \sum_{k=0}^{N-1}\frac{e^{i\alpha_k}+e^{-i\alpha_k}}{2} = 0$$

$$\sum_{k=0}^{N-1}\sin\alpha_k\cos\alpha_k = \sum_{k=0}^{N-1}\left(\frac{e^{i\alpha_k}-e^{-i\alpha_k}}{2i}\right)\left(\frac{e^{i\alpha_k}+e^{-i\alpha_k}}{2}\right) = 0$$





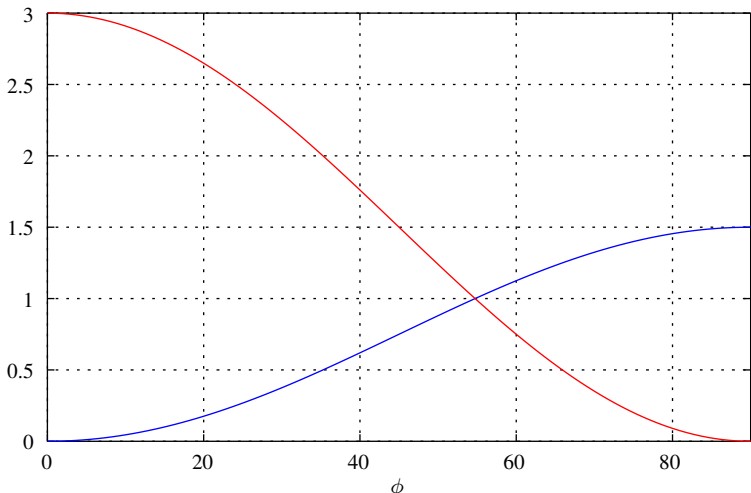

**Figure 2.** Plot of $\frac{N}{2}\sin^2\phi$ (blue) and $N\cos^2\phi$ (red) for $N = 3$.

the matrix $P^T P$ simplifies to

$$P^T P = S = \begin{pmatrix} \frac{N}{2}\sin^2\phi & 0 & 0 \\ 0 & \frac{N}{2}\sin^2\phi & 0 \\ 0 & 0 & N\cos^2\phi \end{pmatrix}. \tag{12}$$

This means that the frame operator $S$ is diagonal for each $\phi$ and $N \geq 3$ with frame bounds, see Figure 2,

$$A = \lambda_{\min} = \min\{\frac{N}{2}\sin^2\phi, N\cos^2\phi\} \text{ and } B = \lambda_{\max} = \max\{\frac{N}{2}\sin^2\phi, N\cos^2\phi\}. \tag{13}$$

5   In the case $A = B$, i.e. $\sin^2\phi = 2\cos^2\phi$ the frame is called tight. The corresponding $\phi$ satisfies then $\sin^2\phi = 2/3$ and hence $\phi_{\text{tight}} = \arcsin\sqrt{\frac{2}{3}} \approx 54.7356$.

The wind retrieval vector is now easily computed as

$$\boldsymbol{v} = S^{-1} P^T \boldsymbol{p} = S^{-1} T \boldsymbol{V}, \tag{14}$$

where

10   $$S^{-1} = \begin{pmatrix} \frac{2}{N}\sin^{-2}\phi & 0 & 0 \\ 0 & \frac{2}{N}\sin^{-2}\phi & 0 \\ 0 & 0 & \frac{1}{N}\cos^{-2}\phi \end{pmatrix} \tag{15}$$

with bounds for $\phi < \phi_{\text{tight}}$: $B^{-1} = \frac{1}{N\cos^2\phi}$ and $A^{-1} = \frac{2}{N\sin^2\phi}$, for $\phi > \phi_{\text{tight}}$: $B^{-1} = \frac{2}{N\sin^2\phi}$ and $A^{-1} = \frac{1}{N\cos^2\phi}$, and for $\phi = \phi_{\text{tight}}$: $A = B = \frac{N}{3}$.



With the help of (14) and (15), the explicit algebraic solution is obtained as

$$
\boldsymbol{v} = \begin{pmatrix} \frac{2}{N\sin^2\phi} & 0 & 0 \\ 0 & \frac{2}{N\sin^2\phi} & 0 \\ 0 & 0 & \frac{1}{N\cos^2\phi} \end{pmatrix} \begin{pmatrix} \sin\alpha_0\sin\phi & \dots & \sin\alpha_{N-1}\sin\phi \\ \cos\alpha_0\sin\phi & \dots & \cos\alpha_{N-1}\sin\phi \\ \cos\phi & \dots & \cos\phi \end{pmatrix} \begin{pmatrix} V_0 \\ \vdots \\ V_{N-1} \end{pmatrix} .
\tag{16}
$$

The matrix multiplications in (16) of the inverse frame operator $S^{-1}$ with the pre-frame operator $T$, whose columns are comprised of the unit vectors describing the beams, yield the explicit solution for the wind vector:

$$
\boldsymbol{v} = \begin{pmatrix} u \\ v \\ w \end{pmatrix} = \begin{pmatrix} \frac{2}{N\sin\phi}\sum_{k=0}^{N-1}\sin\alpha_k V_k \\ \frac{2}{N\sin\phi}\sum_{k=0}^{N-1}\cos\alpha_k V_k \\ \frac{1}{N\cos\phi}\sum_{k=0}^{N-1} V_k \end{pmatrix} .
\tag{17}
$$

### 3.2 Estimation of the retrieval error

Since the wind retrieval for the equispaced VAD sampling case can be explicitly expressed as $\boldsymbol{v} = S^{-1}T\boldsymbol{V}$ it is possible to investigate the propagation of measurement errors in the radial wind components to the final wind vector directly. In what follows, the deterministic as well the stochastic error model will be discussed.

Assume as before, the following deterministic error model $\boldsymbol{V}^\delta = \boldsymbol{V} + \Delta\boldsymbol{V}$, where $\|\Delta\boldsymbol{V}\| \leq \delta$. For the reconstruction error we then obtain $\Delta\boldsymbol{v} = \boldsymbol{v}^\delta - \boldsymbol{v} = S^{-1}T(\boldsymbol{V}^\delta - \boldsymbol{V}) = S^{-1}T\Delta\boldsymbol{V}$, which is

$$
\Delta\boldsymbol{v} = S^{-1} \begin{pmatrix} \sin\phi\sum_{k=0}^{N-1}\sin\alpha_k\Delta V_k \\ \sin\phi\sum_{k=0}^{N-1}\cos\alpha_k\Delta V_k \\ \cos\phi\sum_{k=0}^{N-1}\Delta V_k \end{pmatrix} .
\tag{18}
$$

Therefore, with the help of the Cauchy-Schwartz inequality,

$$
\begin{aligned}
\|\Delta\boldsymbol{v}\|^2 &\leq \|S^{-1}\|^2 \left[ \sin^2\phi\left(\sum_{k=0}^{N-1}\sin\alpha_k\Delta V_k\right)^2 + \sin^2\phi\left(\sum_{k=0}^{N-1}\cos\alpha_k\Delta V_k\right)^2 + \cos^2\phi\left(\sum_{k=0}^{N-1}\Delta V_k\right)^2 \right] \\
&\leq \|S^{-1}\|^2 \left[ \sin^2\phi\frac{N}{2}\|\Delta\boldsymbol{V}\|^2 + \sin^2\phi\frac{N}{2}\|\Delta\boldsymbol{V}\|^2 + \cos^2\phi N\|\Delta\boldsymbol{V}\|^2 \right] \leq A^{-2}N\delta^2 .
\end{aligned}
\tag{19}
$$

Consequently, from (19) we deduce,

$$
\|\Delta\boldsymbol{v}\| \leq A^{-1}\sqrt{N}\delta = \begin{cases} \frac{2\delta}{\sqrt{N}\sin^2\phi} & \text{for} \quad \phi < \phi_{\text{tight}} \\ \frac{3\delta}{\sqrt{N}} & \text{for} \quad \phi = \phi_{\text{tight}} \\ \frac{\delta}{\sqrt{N}\cos^2\phi} & \text{for} \quad \phi > \phi_{\text{tight}} \end{cases} .
\tag{20}
$$

The essential observation in (20) is that an increase of the number of beams leads to a smaller reconstruction error, and that the smallest error (for any $N$) is achieved for $\phi = \phi_{\text{tight}}$, see Figure 3.

Now assume that the measured radial wind components follow the simple stochastic model, $\boldsymbol{V}^\delta = \boldsymbol{V} + \Delta\boldsymbol{V}$, with $\Delta\boldsymbol{V} \sim \mathcal{N}(\boldsymbol{\beta}, \boldsymbol{\Sigma})$, where $\mathcal{N}(\boldsymbol{\beta}, \boldsymbol{\Sigma})$ is the $N$-dimensional normal distribution with expectation vector $\boldsymbol{\beta}$ and variance matrix $\boldsymbol{\Sigma}$. If



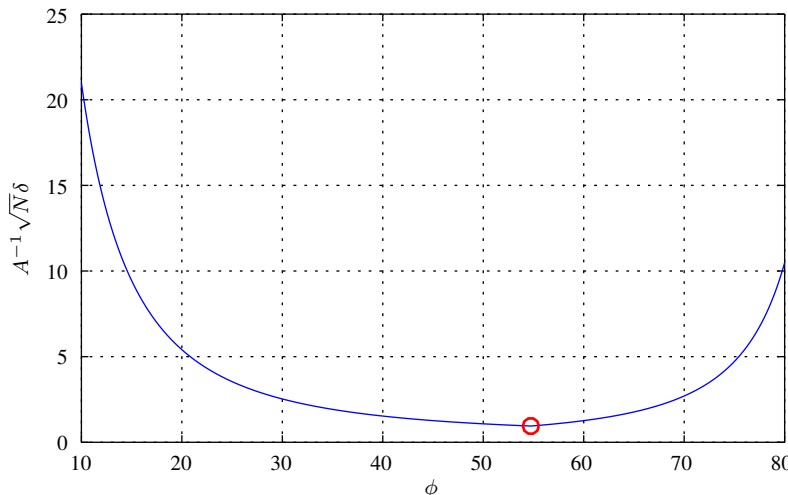

**Figure 3.** Frame bounds in dependence on the angle $\phi$ (for $N = 10$). The red circle indicates the minimum for $\phi_{\text{tight}} = \arcsin\sqrt{2/3}$.

we assume that the components of $\boldsymbol{\beta}$ are constant, $\beta_i = \beta$ for $i = 0, \ldots, N-1$, and $\boldsymbol{\Sigma} = \mathsf{diag}(\sigma^2, \ldots, \sigma^2)$. By computing the expectation of the bias, $\mathsf{E}(\Delta\boldsymbol{v})$, one obtains

$$\mathsf{E}(\Delta\boldsymbol{v}) = \mathsf{E}(S^{-1}T\Delta\boldsymbol{V}) = S^{-1}T\mathsf{E}(\Delta\boldsymbol{V}) = S^{-1}T\boldsymbol{\beta} = S^{-1}\begin{pmatrix} \sin\phi \sum_{k=0}^{N-1} \sin\alpha_k\beta \\ \sin\phi \sum_{k=0}^{N-1} \cos\alpha_k\beta \\ \cos\phi \sum_{k=0}^{N-1} \beta \end{pmatrix} = \frac{\beta}{\cos\phi}\begin{pmatrix} 0 \\ 0 \\ 1 \end{pmatrix}. \tag{21}$$

It can clearly be seen that a constant bias in the radial wind estimates affects only the estimation of the vertical wind
5   component, whereas the horizontal wind vector components remain bias-free. This is due to the symmetry of the sampling which leads to a cancellation of any existing bias in the radial winds.

To compute the mean square error (MSE), observe by standard arguments,

$$\mathsf{E}(\Delta\boldsymbol{v}\Delta\boldsymbol{v}^T) = S^{-1}T\mathsf{E}(\Delta\boldsymbol{V}\Delta\boldsymbol{V}^T)T^T S^{-1} = S^{-1}T(\boldsymbol{\Sigma} + \boldsymbol{\beta}\boldsymbol{\beta}^T)T^T S^{-1} = \mathsf{Var}(\Delta\boldsymbol{v}) + \underbrace{(\mathsf{E}\Delta\boldsymbol{v})(\mathsf{E}\Delta\boldsymbol{v})^T}_{\text{bias}^2},$$

which is clear due to $\boldsymbol{\Sigma}_{jk} = \mathsf{E}(\Delta V_j - \beta)(\Delta V_k - \beta) = \mathsf{E}\Delta V_j \Delta V_k - \beta^2$, which is obvious as by independency it holds for $j \neq k$
10   that $\mathsf{E}\Delta V_j \Delta V_k = \mathsf{E}\Delta V_j \cdot \mathsf{E}\Delta V_k = \beta^2$ and therefore,

$$\mathsf{E}\Delta V_j \Delta V_k = \begin{cases} \sigma^2 + \beta^2, & j = k \\ \beta^2, & j \neq k \end{cases}.$$





In the stochastic regime, the deterministic error estimate can be reproduced. Indeed, it can be observed that,

$$
\begin{aligned}
\mathsf{E}\|\Delta \boldsymbol{v}\|^2 &\leq \|S^{-1}\|^2 \Big(\sin^2\phi \sum_{jk}(\sin\alpha_j \sin\alpha_k + \cos\alpha_j \cos\alpha_k)\mathsf{E}\Delta V_j \Delta V_k + \cos^2\phi \sum_{jk}\mathsf{E}\Delta V_j \Delta V_k\Big) \\
&= \|S^{-1}\|^2 N(\sigma^2+\beta^2) + \|S^{-1}\|^2 \beta^2\Big(\sin^2\phi\big(N\sum_k \cos(2\pi k/N)-N\big)+N(N-1)\cos^2\phi\Big) \\
&= \|S^{-1}\|^2 N(\sigma^2+\beta^2) + \|S^{-1}\|^2 \beta^2 N(N\cos^2\phi-1) \\
&= A^{-2}N\sigma^2 + A^{-2}N^2\beta^2\cos^2\phi \ .
\end{aligned}
$$

This estimate verifies the deterministic recovery error and for growing $N$ this error component can also be made arbitrarly small. The second summand, however, cannot be compensated as it is independent of $N$.

The MSE can be explicitly calculated as follows:

$$
\begin{aligned}
\mathsf{E}\|\Delta \boldsymbol{v}\|^2 &= \mathsf{E}\big((\Delta u)^2 + (\Delta v)^2 + (\Delta w)^2\big) \\
&= \Big(\frac{4}{N^2\sin^2\phi}\sum_{jk}(\sin\alpha_j \sin\alpha_k + \cos\alpha_j \cos\alpha_k)\mathsf{E}\Delta V_j \Delta V_k + \frac{1}{N^2\cos^2\phi}\sum_{jk}\mathsf{E}\Delta V_j \Delta V_k\Big) \\
&= \Big(\frac{4}{N^2\sin^2\phi}N\sigma^2 + \frac{1}{N^2\cos^2\phi}(N\sigma^2+N^2\beta^2)\Big) = \frac{\sigma^2}{N}\Big(\frac{4}{\sin^2\phi}+\frac{1}{\cos^2\phi}\Big) + \frac{\beta^2}{\cos^2\phi} \ .
\end{aligned}
\tag{22}
$$

For $\beta = 0$ and fixed $N$, the choice $\phi = \phi_{\text{tight}}$ yields the smallest value for the MSE. The case $\beta \neq 0$ changes the situation. Let

$$
\mathsf{E}\|\Delta \boldsymbol{v}\|^2 = \frac{\sigma^2}{N}\underbrace{\big(4\sin^{-2}\phi + c\cdot\cos^{-2}\phi\big)}_{=:F(\phi)} \ ,
$$

where $c = 1 + N\frac{\beta^2}{\sigma^2}$. For extremal values, $F'(\phi) = 0$ must be evaluated, which is equivalent to evaluating

$$
(4-c)\sin^4\phi - 8\sin^2\phi + 4 = 0 \ .
$$

For $4-c = 0$ ($\Leftrightarrow N = 3\frac{\sigma^2}{\beta^2}$), the optimal $\phi = \arcsin\sqrt{1/2} = 45°$. In all other cases, the optimal $\phi$ is given by

$$
\sin^2\phi = \frac{4-2\sqrt{c}}{4-c} \iff \phi = \arcsin\sqrt{\frac{4-2\sqrt{c}}{4-c}} \ .
\tag{23}
$$

Formula (23) provides us for each given $N$, $\beta$, and $\sigma$ with an optimal (MSE-minimizing) zenith distance angle $\phi$.

Finally, from the computation of $\mathsf{E}\|\Delta \boldsymbol{v}\|^2$ in (22) it follows that

$$
\mathsf{E}\begin{pmatrix}(\Delta u)^2 \\ (\Delta v)^2 \\ (\Delta w)^2\end{pmatrix} = \begin{pmatrix}\frac{2\sigma^2}{N\sin^2\phi} \\ \frac{2\sigma^2}{N\sin^2\phi} \\ \frac{\sigma^2}{N\cos^2\phi}\end{pmatrix} + \begin{pmatrix}0 \\ 0 \\ \frac{\beta^2}{\cos^2\phi}\end{pmatrix}
\tag{24}
$$

supporting and explaining results obtained by Cheong et al. (2008), who have experimentally shown that the MSE or likewise the RMS error of the wind retrieval is significantly reduced by increasing the number of off-vertical beams in the Doppler beam-swinging technique in the presence of wind field inhomogeneities. Note, however, that for the vertical wind component only the random error can be reduced by an increase of $N$.



## 4 Conclusions

In this note, the mathematical concept of frames is applied to the analysis of the spatial (beam configuration) sampling set-up for Doppler profilers for the case of a horizontally homogeneous and stationary wind field. It could be shown that it is possible to derive a compact explicit least-square wind retrieval solution for a typical symmetric VAD scanning scheme. Such an explicit

5 formula was hitherto not published yet. Besides its simplicity, it allows for a straightforward stability analysis in the practically relevant case of noisy data. The explicit solution exhibits the known fact that the VAD-based estimate for the horizontal wind components is unbiased even if the radial wind components have a constant (direction-independent) bias. Furthermore it was shown that the MSE retrieval error is $\propto 1/N$ up to a constant offset due to the bias, which means that a larger number of off-zenith beam directions is beneficial to reduce the variance of the wind vector components. The total retrieval error is depending

10 on the zenith distance $\phi$. For the most relevant case $\beta = 0$ it is minimal if the beam vectors form a tight frame. The optimal beam zenith distance angle for this case was calculated as $\phi = \arcsin \sqrt{\frac{2}{3}} = 54.7356°$. It must be noted that this elevation angle of $35.264°$ is a much lower value than what is used in practical configurations of most Doppler systems. The reason is the conflicting requirement to keep the sampled volume small, in an attempt to minimize deviations from a constant wind field. Technical constraints like the usable Nyquist velocity range and limited scanning capabilities of phased array antennas

15 are additional reasons why the theoretically optimal elevation is so far not used in practice.





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
