# Peer review of "Mean wind vector estimation using the Velocity-Azimuth-Display (VAD) method: An explicit algebraic solution"

_Atmospheric Measurement Techniques, 2016_

## Referee Comment (RC1) · Anonymous Referee #1 · 3 Feb 2017

General comments:

The paper deals with a well-known practical issue of determining a vertical wind profile from a set of known radar measurements. The authors are able by using the measurement directions as a frame to provide a new derivation and an explicit solution (17). The outcome is similar to a Fourier transform of the equidistant radial velocities. The results that the vertical and horizontal winds are the first Fourier components of the azimuthal wind field is well-known although the Fourier expansion is not usually presented explicitly (but see Browning Wexler, page 107). Hence the analytical result is not really new. The greatest value of the derivation is that is provides a starting point to the error and stability analysis which is most interesting and provides new results. The frame

concept is very efficient in this respect and allows the authors to derive and optimal angle for the measurement. The elevation angle (35 deg) is larger than those recommended for weather radars, to avoid fall speed contamination in rain, and smaller than those usually used for wind profiling radars as the authors discuss. It would be a valuable addition to estimate how much the error increases when the typical angles for the radar systems are used instead of the optimal angle, assuming that the assumptions are valid.

I disagree with the finding of the authors that increasing the number of beam directions reduces the error. If the number of beams is increased, but the statistical error of radial velocities ($\sigma$) is kept constant, the measurement time is increased, which surely decreases the random error. In case the measurement time is kept constant and number of beams increased, the number of observation along any radial is decreased and the error of radial velocities increased. Hence no gain is seen in the derived winds. The error of the wind components depends on the measurement time, independent on how the measurement is arranged, assuming that the problems stays reasonably well-conditioned.

Specific comments:

Page 4, line 11: Vector p is used here although it is introduced only a few lines later

Page 5, line 15 The paper is more mathematical in nature than typical papers in AMT. The authors have taken the readership into account rather well. But I just wonder if introducing $(T^*)^T$ for Eq.(8) is completely necessary.

Page 5, lines 20-23 appear unnecessarily complicated. The last equation should have a number. Section 3.2 This section is less organized as the rest of the paper. I suggest presenting only the stochastic case. In case both error cases are treated the authors should consider the notations. In the deterministic case the error (delta) is given for the full vector, whereas in the stochastic case they are component wise. This might confuse the readers.

Page 10, lines 13-17. There are many ways to solve the angle, but to me it appears much simpler to solve for the $tan^2\phi = 2/\sqrt{c}$, without a need to solve for the case c=4 separately.

---

## Referee Comment (RC2) · Anonymous Referee #2 · 15 Feb 2017

The authors have addressed my original criticisms, comments and questions. Just a couple of minor suggestions:

Conclusions section, lines 9-10: Change "The total retrieval error is depending on ..." to the "The total retrieval error is dependent upon..."

Also in several places the author makes reference to the "zenith distance" or the "zenith distance angle." I would suggest changing this to simply "zenith angle".

---

## Author Comment (AC1)

Reply to Reviewer comments on Manuscript doi:10.5194/amt-2016-365-RC1,2017 Mean wind vector estimation using the Velocity-Azimuth-Display (VAD) method: An explicit algebraic solution by G. Teschke and V. Lehmann

**12.05.2017**

**1 Reviewer #1**

**1.1** General comments**

The paper deals with a well-known practical issue of determining a vertical wind profile from a set of known radar measurements. The authors are able by using the measurement directions as a frame to provide a new derivation and an explicit solution (17). The outcome is similar to a Fourier transform of the equidistant radial velocities. The results that the vertical and horizontal winds are the first Fourier components of the azimuthal wind field is wellknown although the Fourier expansion is not usually presented explicitly (but see Browning Wexler, page 107). Hence the analytical result is not really new.

We thank the reviewer for pointing out that an example for the explicit solution for the wind vector retrieval in our equation (17) is given in [Browning and Wexler(1968)], namely between their equations (7) and (8). However, these authors have not mentioned the rather general significance of these formulae since they appear only for a special case and are neither discussed nor numbered. We can only speculate whether or not Browning and Wexler fully appreciated the importance of their example and would therefore respectfully disagree with the referee that equation (17) in our paper is well-known. The derivation of an explicit formula has, to the best of our knowledge, not been published so far and neither appears in standard textbooks on radar meteorology, like [Doviak and Zrnić(1993)], nor in similar literature about lidar, e.g. [Weitkamp(2005)]. By the same token, a formula as our equation (24) is given without any proof or reference on page 497 in [Henderson et al.(2005)Henderson, Gatt, Rees, and Huffaker], however without the term due to a possible bias in the radial wind.

General explanations of the VAD method typically mention *the fitting of a sinusoidal curve to the data* which seems to indicate that the solution is predominantly obtained through numerical methods in practice. We believe that it is therefore fully justified to publish the derivation of the general algebraic solution to the least square problem in the case of symmetric sampling as well as the error propagation results for this case.

The greatest value of the derivation is that is provides a starting point to the error and stability analysis which is most interesting and provides new results. The frame concept is very efficient in this respect and allows the authors to derive and optimal angle for the measurement. The elevation angle (35 deg) is larger than those recommended for weather radars, to avoid fall speed contamination in rain, and smaller than those usually used for wind profiling radars as the authors discuss. It would be a valuable addition to estimate how much the error increases when the typical angles for the radar systems are used instead of the optimal angle, assuming that the assumptions are valid.

The question of an optimal elevation angle for VAD-like wind vector retrievals has been discussed for weather radars, radar wind profilers and lidars, see e.g. [Röttger and Larsen(1990)]. Essentially, there are reasons to choose the zenith distance angle as small as possible, most importantly to assuring a better homogeneity of the mean wind as well as reasons to use zenith distance angles as large as possible, to restrict geometrical effects in the error propagation from the radial wind measurement onto the wind vector components.

It is important to appreciate that for the practically relevant case of an estimate of only the horizontal components of the mean wind (since the mean vertical wind component will mainly be close to zero, except for special meteorological conditions) no such optimum can be derived from purely geometrical arguments. For clarification, we have added this remark to the paper.

Any discussion of optimal sampling configurations for practical systems must take system characteristics into account, like aspect-sensitivity for VHF radars, antenna radiation pattern restrictions and loss of sensitivity for phased arrays with fixed orientation, hydrometeor contamination for weather radars or mechanical limitations for simple optical scanners used in lidars. The paper therefore does not attempt to provide a recipe for setting-up wind measuring remote sensing instruments, but provides only the mathematical foundation for the geometric aspect of the sampling.

I disagree with the finding of the authors that increasing the number of beam directions reduces the error. If the number of beams is increased, but the statistical error of radial velocities ( $\sigma$ ) is kept constant, the measurement time is increased, which surely decreases the random error. In case the measurement time is kept constant and number of beams increased, the number of observation along any radial is decreased and the error of radial velocities increased. Hence no gain is seen in the derived winds. The error of the wind components depends on the measurement time, independent on how the measurement is arranged, assuming that the problems stays reasonably well-conditioned.

We are afraid that the question posed by the reviewer is out of the scope of this short paper and would deserve further investigations, because the problem is not as simple as it may seem:

The line of thought rests on the assertion that an increase of measurement or dwell time for a single beam direction decreases the random error linearly with time and vice versa. Since the Doppler velocity is derived through a spectral analysis of the receiver signal it is clear that the attainable frequency resolution is inversely proportional to the dwell time. However, the received signal in every remote sensing instrument always has a random component due to various sources of noise and is furthermore influenced by the degree of stationarity of the physical scattering process, so the attainable accuracy has no simple linear dependence on observation time. In practice it is even possible that the random error can increase with increasing dwell time if transient clutter phenomena are not properly suppressed in radar or if the scattering process is nonstationary.

**1.2** Specific comments**

Page 4, line 11: Vector p is used here although it is introduced only a few lines later

The referee is right. It is corrected accordingly.

Page 5, line 15 The paper is more mathematical in nature than typical papers in AMT. The authors have taken the readership into account rather well.

But I just wonder if introducing  $(T^*)^T$  for Eq.(8) is completely necessary. The referee is right. We have changed this part.

Page 5, lines 20-23 appear unnecessarily complicated. The last equation should have a number. Section 3.2 This section is less organized as the rest of the paper. I suggest presenting only the stochastic case. In case both error cases are treated the authors should consider the notations. In the deterministic case the error (delta) is given for the full vector, whereas in the stochastic case they are component wise. This might confuse the readers.

Page 5, lines 20-23: we have condensed the deduction and gave a number to this formula. In section 3.2. we present both scenarios. The notation should not confuse the reader: vectors are written in bold letters  $\Delta \mathbf{v}$ ,  $\Delta \mathbf{V}$ and vector components are written in non-bold letters  $\Delta u$ ,  $\Delta v$ ,  $\Delta w$ .

Page 10, lines 13-17. There are many ways to solve the angle, but to me it appears much simpler to solve for the  $\tan^2(\Phi) = 2/\sqrt{c}$ , without a need to solve for the case c=4 separately.

The referee is right, there are many ways to solve the angle. With a simple trigonometric computation your result is obtained. We have condensed our illustration accordingly.

**2 Reviewer #2**

The authors have addressed my original criticisms, comments and questions. Just a couple of minor suggestions: Conclusions section, lines 9-10: Change "The total retrieval error is depending on ..." to the "The total retrieval error is dependent upon..." Also in several places the author makes reference to the "zenith distance" or the "zenith distance angle." I would suggest changing this to simply "zenith angle".

We thank the reviewer for these hints and have incorporated them into the paper.

**References**

[Browning and Wexler(1968)] Browning, K. and Wexler, R.: The Determination of Kinematic Properties of a Wind Field Using Doppler Radar, J. Appl. Meteor., 7, 105–113, 1968.

- [Doviak and Zrnić(1993)] Doviak, R. J. and Zrnić, D. S.: Doppler Radar and Weather Observations, Academic Press, 1993.
- [Henderson et al.(2005)Henderson, Gatt, Rees, and Huffaker] Henderson, S. W., Gatt, P., Rees, D., and Huffaker, R. M.: Wind Lidar, in: Laser Remote Sensing, edited by Fujii, T. and Fukuchi, T., CRC Press, 2005.
- [Röttger and Larsen(1990)] Röttger, J. and Larsen, M.: UHF/VHF Radar Techniques for Atmospheric Research and Wind Profiler Appications, in: Radar in Meteorology, edited by Atlas, D., Amer. Meteor. Soc., 1990.
- [Weitkamp(2005)] Weitkamp, C.: Lidar: range resolved optical remote sensing of the atmosphere, Springer, 2005.